# The DEXA-CORT trial: study protocol of a randomised placebo-controlled trial of hydrocortisone in patients with brain tumour on the prevention of neuropsychiatric adverse effects caused by perioperative dexamethasone

Anne-Sophie C A M Koning [1], Djaina D Satoer,[2] Christiaan H Vinkers [3,4] Amir H Zamanipoor Najafabadi [5] Nienke R Biermasz [6] Rishi D S Nandoe Tewarie,[5] Wouter A Moojen,[5] Elisabeth F C van Rossum [7] Clemens M F Dirven,[2] Alberto M Pereira [1] Wouter R van Furth [5] Onno C Meijer [1]

AMP and WRvF contributed equally.

For numbered affiliations see end of article.

**Correspondence to**
Anne-Sophie C A M Koning;
a.c.a.koning@lumc.nl

## ABSTRACT

**Introduction** The synthetic glucocorticoid dexamethasone can induce serious neuropsychiatric adverse effects. Dexamethasone activates the glucocorticoid receptor (GR) but, unlike endogenous cortisol, not the mineralocorticoid receptor (MR). Moreover, dexamethasone suppresses cortisol production, thereby eliminating its MR binding. Consequently, GR overactivation combined with MR underactivation may contribute to the neuropsychiatric adverse effects of dexamethasone. The DEXA-CORT trial aims to reactivate the MR using cortisol to reduce neuropsychiatric adverse effects of dexamethasone treatment.

**Methods and analysis** The DEXA-CORT study is a multicentre, randomised, double-blind, placebo-controlled trial in adult patients who undergo elective brain tumour resection treated perioperatively with high doses of dexamethasone to minimise cerebral oedema. 180 patients are randomised between treatment with either two times per day 10 mg hydrocortisone or placebo during dexamethasone treatment. The primary study outcome is the difference in proportion of patients scoring ≥3 points on at least one of the Brief Psychiatric Rating Scale (BPRS) questions 5 days postoperatively or earlier at discharge. Secondary outcomes are neuropsychiatric symptoms, quality of sleep, health-related quality of life and neurocognitive functioning at several time points postoperatively. Furthermore, neuropsychiatric history, serious adverse events, prescribed (psychiatric) medication and referrals or evaluations of psychiatrist/psychologist and laboratory measurements are assessed.

**Ethics and dissemination** The study protocol has been approved by the Medical Research Ethics Committee of the Leiden University Medical Center, and by the Dutch competent authority, and by the Institutional Review Boards of the participating sites. It is an investigator-initiated study with financial support by The Netherlands Organisation for Health Research and Development

### Strengths and limitations of this study

► The DEXA-CORT study is the first double-blind, placebo-controlled intervention trial in patients with a brain tumour aiming to reduce neuropsychiatric adverse effects of perioperative dexamethasone.
► Use of validated outcome measurements.
► Evaluation of acute and potential long-term effects of the intervention.
► Duration of dexamethasone treatment differs between participants.
► Not all patients experience adverse effects following dexamethasone treatment, and patients without adverse effects are included.

(ZonMw) and the Dutch Brain Foundation. Results of the study will be submitted for publication in a peer-reviewed journal.

**Trial registration number** NL6726 (Netherlands Trial Register); open for patient inclusion. EudraCT number 2017-003705-17.

## INTRODUCTION

Synthetic glucocorticoids (GCs) are widely used across many different clinical disciplines, in particular, immunology, pulmonology, neurosurgery and oncology. Their potent anti-inflammatory and immunosuppressive effects and ability to control cerebral oedema ensure effective treatment.[1–3] However, moderate to high doses of synthetic GCs frequently cause severe side effects.[4] These include serious neuropsychiatric conditions such as sleep disturbances, delirium, depression, mania, psychosis and suicidality.[5] Despite

these mentioned problems, dexamethasone is a standard treatment in patients with a brain tumour for more than 50 years since no better alternatives are present.[6–8] Evidently, these adverse effects negatively affect well-being and quality of life. In particular, depressive symptoms, a common problem in patients with a brain tumour, appeared to be an independent predictor for quality of life rating.[9] The risks of glucocorticoid-induced neuropsychiatric adverse effects in this and other patient groups are, therefore, serious.[10] At present, there is no preventive treatment for these unwanted and serious neuropsychiatric adverse effects.

The naturally produced hormone cortisol acts on both the glucocorticoid receptor (GR) and mineralocorticoid receptor (MR), whereas the synthetic GC dexamethasone mainly activates GR with low or absent binding to the MR in vivo.[11] The increased GR activity is mediating the therapeutic goal of the treatment, that is, reduction of oedema. However, dexamethasone strongly suppresses endogenous cortisol production through GR-mediated negative feedback mechanisms in the hypothalamus–pituitary–adrenal (HPA) axis.[12 13] The MR is also a prominent cortisol receptor in the brain, and its activation by cortisol binding strongly decreases after the HPA axis suppression.[14] While low concentrations of dexamethasone poorly penetrate the blood brain barrier,[15] higher clinically used doses do reach the brain. This has led to the hypothesis that, in addition to the strongly stimulated GR, reduced brain MR activity may underlie the

neuropsychiatric adverse effects during dexamethasone treatment.[16]

MR activity has indeed been related to psychopathology. Correlatively, decreased MR expression has been observed in several psychiatric disorders.[17] Functionally, both pharmacological intervention studies and a genetic gain-of-function variant of MR seem to have protective effects.[18–20] Therefore, we hypothesise that increasing MR activity with physiological doses of cortisol (=hydrocortisone) may prevent or substantially diminish the disrupting neuropsychiatric effects in patients with a brain tumour who receive high doses of perioperative dexamethasone, see figure 1. Such a positive effect of adding hydrocortisone was previously found in a subgroup of children who experienced severe adverse effects after dexamethasone treatment for childhood leukaemia.[21] Therefore, the aim of the present study is to investigate the effects of hydrocortisone replacement in adult patients with a brain tumour, compared with placebo, as add-on to the treatment with dexamethasone, on the development of neuropsychiatric adverse effects that can be caused by dexamethasone.

## METHODS AND ANALYSIS
This manuscript is written according to Standard Protocol Items: Recommendations for Interventional Trials (online supplemental file S1).[22 23]

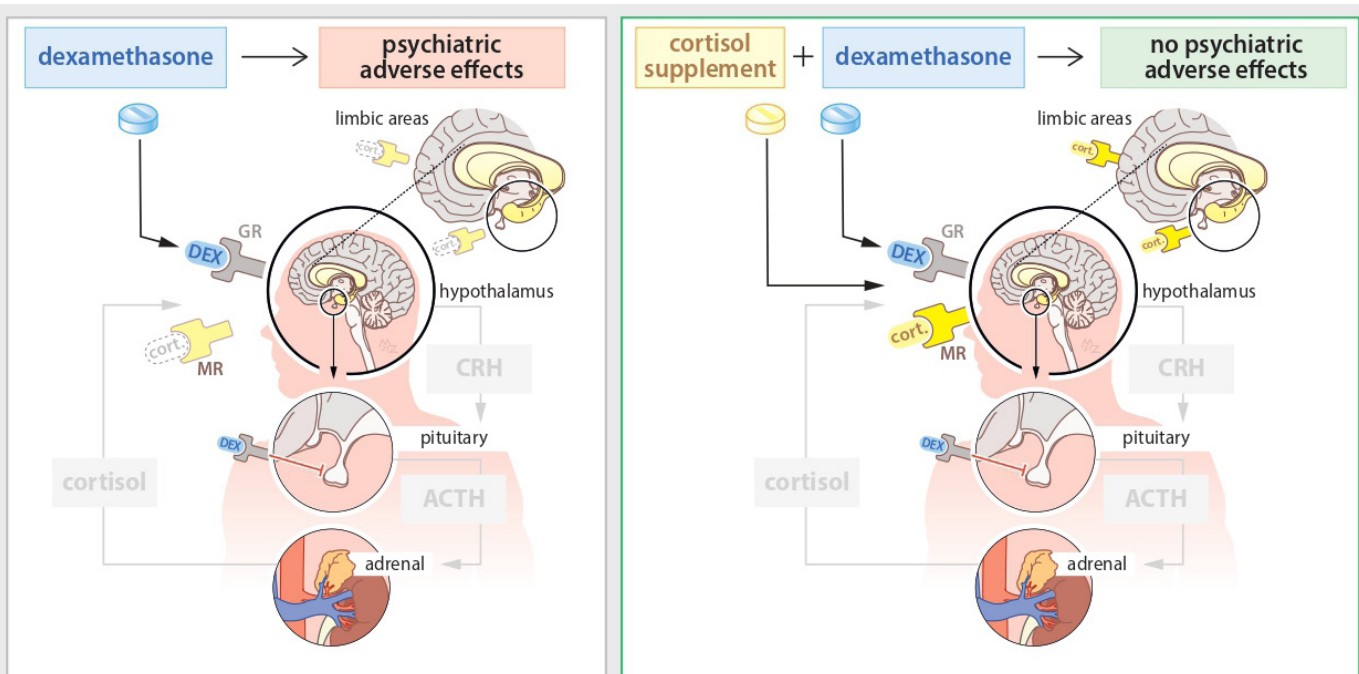

**Figure 1** Preventing neuropsychiatric adverse effects of dexamethasone with co-administration of cortisol. Dexamethasone suppresses the HPA-axis and thereby the adrenocortical cortisol production, leaving cortisol-preferring MRs unoccupied. Re-activating the MR using cortisol might reduce or prevent neuropsychiatric adverse effects of dexamethasone treatment via activation of brain MR. ACTH, adrenocorticotropic hormone; CRH, corticotrophin-releasing hormone; GR, glucocorticoid receptor; HPA, hypothalamus–pituitary–adrenal; MR, mineralocorticoid receptor.

## Study design and setting

The DEXA-CORT study is a multicentre, randomised, double-blind, placebo-controlled trial investigating the effect of hydrocortisone replacement as cotreatment to reduce perioperative dexamethasone-related neuropsychiatric adverse effects in patients with brain tumours undergoing elective neurosurgery. A total of 180 patients will be randomised in a 1:1 ratio, stratified per study centre and on tumour type. With randomisation, we intend to equally distribute factors that can have an influence on our main outcome; neuropsychiatric effects, for example, the underlying pathology, the location of the tumour, extent of the surgery and psychiatric history. The participating centres are Leiden University Medical Center (LUMC), Erasmus University Medical Center (EMC), Haaglanden Medical Center (HMC) and University Medical Center Utrecht (UMCU). The primary outcome measure is the proportion of patients with neuropsychiatric symptoms 5 days after surgery — or earlier if discharge is within 5 days — compared between the hydrocortisone and placebo-treated group. The study is investigator-initiated with the LUMC as sponsor and is funded by ZonMw and the Dutch Brain Foundation.

## Eligibility criteria

Adult patients (≥18 y) diagnosed with a brain tumour (glioma or meningioma) and undergoing elective surgical resection are eligible for inclusion. All patients receive high-dose dexamethasone according to institutional protocols. Since perioperative dexamethasone dosing schemes differ among the participating centres, a minimal perioperative cumulative dexamethasone dose of 24 mg or more in 6 days is required. Initially, preoperative dexamethasone of more than 1 day was considered an exclusion criterium. During the course of the study, it appeared that many patients, especially those with malignant tumours and severe oedema, had dexamethasone prescribed several days before the planned surgery. Consequently, many patients were not eligible for study participation, threatening completion of inclusion. Hence, this exclusion criterium was discarded with consent of the Data Safety Monitoring Board (DSMB). Detailed inclusion and exclusion criteria are presented in figure 2.

## Study outline

A timeline of the study is shown in figure 3. Eligible patients are approached in an outpatient setting by specialised nurses in consultation with the treating neurosurgeon. The study is explained and written information is provided (online supplemental files S2,3). After written informed consent by the coordinating investigator or principle investigators, patient characteristics (age, gender, educational level, medical background, medication use, tumour type and tumour location) are recorded. The informed consent form includes consent for collection for blood sample and residual tumour tissue. Before start

**Inclusion criteria:**

- Age = ≥18 years
- Cranial glioma or meningioma patients scheduled to undergo surgery (resection)
- Minimal dose of peri-operative cumulative dexamethasone exposure of 24mg or more in 6 days
- Good clinical condition; KPS ≥70
- Life expectancy ≥ 6 months

**Exclusion criteria:**

- Non-native speakers of Dutch or insufficient command of the Dutch language
- Patients that are unable to overview consequences of trial participation
- Patients with severe aphasia
- Patients that are not able to fill in the questionnaires because of cognitive impairments at the discretion of the physician
- Patients with psychiatric diseases or neurological deficits that interfere with the study to the judgement of treating physician
- Patients with prior diagnosis of secondary adrenal insufficiency/already taking hydrocortisone

**Figure 2** Eligibility criteria of the DEXA-CORT study. KPS, Karnofsky Performance Status.

of intervention, a baseline study visit is planned that includes a structured psychiatric interview, different self-report questionnaires, and neurocognitive tests. A selection of the questionnaires can be filled out online via the web-based database programme Castor Electronic Data Capture (EDC; Castor, Amsterdam, The Netherlands). The intervention starts on the day of hospital admission, concurrent with the dexamethasone treatment 1 day prior to planned surgery. 5 days postoperatively – or earlier at discharge – a structured psychiatric interview and neurocognitive screening are conducted, and questionnaires are filled out. The study medication is provided to the patients according

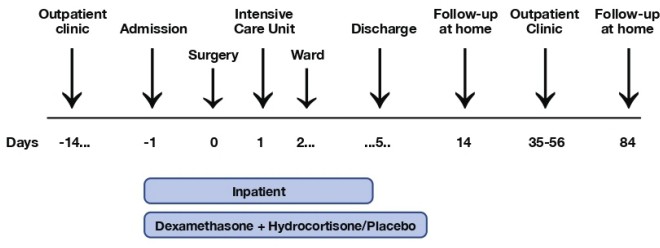

**Figure 3** The DEXA-CORT study timeline.

to clinical care and is continued as long as dexamethasone is prescribed. A follow-up visit is planned 5 to 8 weeks postoperatively coinciding with the regular care assessment, at which a structured psychiatric interview and neurocognitive tests are performed and different questionnaires are filled out. Study visits take place at the medical centre of treatment. 2 weeks and 3 months postoperatively questionnaires can be filled out online via Castor EDC. Study duration from start of intervention until completion of the last questionnaires is 3 months. The study schedule is shown in table 1. In case the baseline study visit is planned the day before surgery, no neurocognitive tests will be performed – instead a neurocognitive screening may be performed – and presentation of the questionnaires is depend on the patients' psychological and physical condition. To improve adherence the follow-up visits are coinciding with the regular care assessments.

**Table 1** The DEXA-CORT study schedule of enrolment, intervention and assessments

| | | Study period | | | | | | | |
|---|---|---|---|---|---|---|---|---|---|
| | Outpatient clinic | Baseline | Treatment | | | | Follow-up | Follow-up | Close-out |
| | | | Admission | Surgery | Ward | Discharge | | | |
| Days | | −14 – −1 | −1 | 0 | 1–4 | 3–5 | 14 | 35–56 | 84 |
| Visits | V$_0$ | V$_1$ | V$_1$/V$_2$ | | | | | V$_2$/V$_3$ | |
| **Enrolment** | | | | | | | | | |
| Eligibility screen | X | | | | | | | | |
| Informed consent | | X | X | | | | | | |
| Baseline characteristics | | X | X | | | | | | |
| Allocation | | X | X | | | | | | |
| **Interventions** | | | | | | | | | |
| Hydrocortisone | | | X | | | →————————→ | | | |
| Placebo | | | X | | | →————————→ | | | |
| **Assessments** | | | | | | | | | |
| BPRS | | X | X | | | X | | X | |
| Mini-screen | | X | X | | | | | | |
| HADS | | X | (X) | | | | X | | X |
| PANAS | | X | (X) | | (X) | X | X | X | X |
| ASRM | | X | (X) | | | | X | | X |
| NPI-Q | | X | X | | | X | | X | |
| DOS | | | | | X | | | | |
| LSEQ | | X | (X) | | | X | X | X | X |
| QoL-C30/BN20 | | X | (X) | | | | | | X |
| Cognitive tests | | X | | | | | | X | |
| MoCA | | | (X) | | | X | | | |
| ABC | | X | X | | | X | | | |
| SAEs | | | X | | | →————————→ | | | |
| AEs | | X | ←————————————————————————————————————→ | | | | | | X |
| Medication | | X | ←————————————————————————————————————→ | | | | | | X |
| Psychiatric referrals | | X | ←————————————————————————————————————→ | | | | | | X |
| **Laboratory measures** | | | | | | | | | |
| Tube of blood | | | | X | | | | | |
| Residual tumour tissue | | | | X | | | | | |

ABC, Aphasia Bedside Check; ASRM, Altman Self-Rating Mania Scale; BPRS, Brief Psychiatric Rating Scale; DOS, Delirium Observation Screening; HADS, Hospital Anxiety and Depression Scale; LSEQ, Leeds Sleep Evaluation Questionnaire; MoCA, Montreal Cognitive Assessment; NPI-Q, Neuropsychiatric Inventory Question; PANAS, Positive and Negative Affect Schedule; QoL, Quality of Life questionnaires; (S)AEs, (serious) adverse events.

## Intervention

Patients will be treated according to routine clinical care, based on local treatment protocols. The investigational product is hydrocortisone with placebo as comparator, both produced by Tiofarma B.V., Oud-Beijerland, The Netherlands. The Clinical Pharmacy and Toxicology department of the LUMC will repack, label and randomise the study medication. 180 patients are randomised between treatment with either twice daily 10 mg hydrocortisone or placebo, both orally administered concurrent with the full course of perioperative dexamethasone treatment; mostly with a dexamethasone taper schedule. Study medication is available for 20 days; assuming the slow dexamethasone taper schedule is used, treatment will continue on average for 15 days. In the event that dexamethasone treatment re-starts after completion of its taper schedule, study medication will not be available. The study medication is prescribed by the physician and administration details are recorded. Treatment dosage is targeted towards physiological cortisol replacement. During hospitalisation, the hydrocortisone or placebo is administered in the morning at 08:00 hours and in the evening between 18:00 hours and 20:00 hours, because at these times dexamethasone is administered according to local standard procedures. To minimise the number of errors in medicine administrations we decided to adhere to the existing logistics. When participants go home and have to taper the dexamethasone, they receive a taper schedule from the nurses. Depending on the schedule, they receive further instructions on the hydrocortisone/placebo administration (both verbally and on paper). For manageability reasons we decided to always take the hydrocortisone/placebo at the first and the last time point of the dexamethasone administration. Study medication and information is handed out to the patients by the nurses, in consultation with the coordinating investigator. Patients will receive the amount of study medication they need to minimise medication errors. Drug accountability is checked by the coordinating investigator.

## Allocation, randomisation and blinding

Randomisation is controlled by the Clinical Pharmacy and Toxicology department of LUMC using block randomisation in a 1:1 ratio stratified for each medical centre participating in the study, and for tumour type (glioma or meningioma) with block size of 10. New study participants are assigned to a randomisation number by the coordinating investigator. The coordinating investigator will contact the local pharmacy about the patient and its randomisation number. The local pharmacy assigns the randomisation number to a medication number and then the pharmacists know the treatment allocation. The randomisation list is stored at the department and will not be disclosed until all participants have completed the trial. Only in

case of emergency and the knowledge of the assigned treatment is required, the randomisation code can be broken. The study is blinded for everyone involved, except the pharmacy who allocates participants to treatment. In addition, hydrocortisone and placebo pills are identical in terms of appearance and taste ensuring blinding.

## Assessments during the study

Assessments during the study include neuropsychiatric symptoms, quality of sleep, neurocognitive assessment, registration of (serious) adverse events ((S) AEs), prescribed medication, and referrals to psychiatrist/psychologist. We plan to evaluate potential long term follow-up effects on health-related quality of life (HRQoL). However, we did not include a specific measure for PTSD, even if a 'planned critical illness' like elective brain surgery may induce this disease.[24] Furthermore, blood and residual tissue from tumour specimens will be collected to verify MR targeting and molecular effects of hydrocortisone cotreatment.

### Neuropsychiatric symptoms
*Brief Psychiatric Rating Scale*
Neuropsychiatric symptoms will be measured with the Brief Psychiatric Rating Scale (BPRS) V.4.0.[25] The BPRS is an interview, which provides a description of major symptom characteristics in psychiatric patients. The BPRS covers the presence and severity of 24 symptoms scored on a 7 point scale (1=not present, 7=extremely severe). Possible scores vary from 24 to 168. The BPRS will be used for our primary outcome.

*Mini screen*
A mini screen will be used to determine the patient's neuropsychiatric history. This mini screen comprises 22 'yes' or 'no' questions about different neuropsychiatric symptoms.[26]

*Hospital Anxiety and Depression Scale*
To determine the levels of anxiety and depression in patients, the Hospital Anxiety and Depression Scale (HADS) questionnaire will be used.[27] The HADS comprises 14 items; 7 related to anxiety and 7 to depression. Each question has 4 different answer options, ranging from 0 to 3. Patients scoring 8–10 on the anxiety or depression domain are classified as possibly suffering from anxiety or depression, respectively, and a score ≥11 as definite anxiety or depression.

*Positive and Negative Affect Schedule*
To determine the positive and negative affect (NA) of patients, the Positive and Negative Affect Schedule (PANAS) is used. The PANAS comprises a 20-item mood scale yielding two subscales; positive affect (PA) and NA.[28] Patients rate how they felt 'in general' on a 5-point scale; 1=very slightly or not at all, 2=a little, 3=moderately, 4=quite a bit and 5=very much. PA

scores range from 10 to 50, with higher scores representing higher levels of PA. NA scores range from 10 to 50, with lower scores representing lower levels of NA.

### The Altman Self-Rating Mania Scale

The Altman Self-Rating Mania (ASRM) Scale is used to assess the presence and/or severity of manic symptoms.[29] The ASRM consists of 5-items consisting of five statements. At the 5 items, the patient chooses one statement that best describes the way he or she has been feeling for the past week.

### Neuropsychiatric Inventory Question

The Neuropsychiatric Inventory Question (NPI-Q) is a questionnaire to be filled out by a family member or care taker of the patient, that specifically assesses neuropsychiatric symptoms of the patient in the last month.[30–32] The NPI-Q consists of 12 domains of symptoms. Every domain needs to be answered with 'yes' (is present) or 'no' (is absent). When the symptom is present, the partner rates the severity of the symptom (3-point scale) and the psychiatric burden (6-point scale) for him/herself. The NPI-Q gives a total score for the severity (range 0–36) and a total score for emotional burden (range 0–60).

### Delirium Observation Screening

The Delirium Observation Screening (DOS) scale is used to determine the presence of delirium in patients and includes 13 observation items of behaviour that shows symptoms of delirium.[33] Nurses measure the different items during the day, evening and night. Each item has three answer options; 'never', 'sometimes—always' and 'not known'. These three scores are then scored with 0, 1 or no points. The total DOS scale score is the sum of the total DOS score at day, evening and night divided by 3. Patients scoring ≥3 are probably suffering from delirium.

### Quality of sleep

To determine the sleep quality, the Leeds Sleep Evaluation Questionnaire (LSEQ) is used.[34 35] LSEQ contact information and permission to use is available from Mapi Research Trust, Lyon, France, https://eprovidemapi-trustorg. LSEQ is a reliable instrument to assess sleep quality changes during psychopharmacological treatment intervention. This questionnaire is a self-report measure with 10-items consisting of a 10-cm-line visual analogue rating scale that evaluates four domains; ease of initiating sleep, quality of sleep, ease of waking and behaviour following wakefulness. The LSEQs 10-cm-horizontal line has two extreme states defined at the end of the line (eg, tired=score of 0, alert=score of 10). Patients respond by placing a vertical mark on the line. Each question is scored out of 10 (eg, 0 cm=score 0, 5 cm=score of 5).

### Health-related quality of life

HRQoL will be measured with the European Organisation for Research and Treatment of Cancer(EORTC) core quality of life questionnaire (QLQ-C30) complemented with the QLQ-BN20.[36 37] The QLQ-C30 is specific in assessing the HRQoL of patients with cancer participating in clinical trials. It consists of 30 items, of which 28 have a 4-point scale (1=not at all, 4=very much). Two questions have a 7-point scale (1=very poor, 7=excellent). A high score represents a higher response level; high/healthy level of functioning, high QoL and a high level of symptoms. The QLQ-BN20 is a validated module specific for patients with brain neoplasms. It includes 20 items with a 4-point scale (1=not at all, 4=very much), and the higher the score, the more symptoms or problems.

### Neurocognitive assessment

To assess the cognitive domains verbal memory, executive functioning and attention, we used the following standardised neurocognitive tests: Hopkins Verbal Learning Test–Revised—Free Recall,[38] Trail Making Test A and B,[39] Controlled Oral Word Association[40] and Medical Outcome Study (MOS) cognitive functioning scale.[41] Raw scores can be transformed into z-scores in order to compare patients' performance to control participants (and can be corrected for demographic variables, such as age and education).

The Montreal Cognitive Assessment (MoCA) is a cognitive screening instrument for mild cognitive dysfunction.[42] It measures different cognitive domains; attention and concentration, executive functions, memory, language, visuoconstructional skills, conceptual thinking, calculations and orientation. The total score is 30 points; a score of ≥26 is a normal score.

The Aphasia Bedside Check (ABC) is a 10 minute screening tool that globally determines the presence of aphasia with 7 items for language comprehension and 7 items for language production.[43] This screening will only be performed in patients who have a left-sided brain tumour with (hetero-) anamnestic language problems. With ≥3 mistakes, an aphasia may be present and the patient should not be included in the study.

### SAEs—medication—referral to psychiatrist/psychologist

(S)AEs, prescribed medication (especially psychiatric medication) and referrals or evaluations of psychiatrist/ psychologist will be recorded.

### Laboratory measures

We try to collect 1 tube of blood taken during anaesthetic procedures for surgery. This will be used for DNA analyses to determine MR haplotype. Additionally, cortisol and glucose levels will be measured in blood.

If possible during the surgical procedure tumour tissue and peritumoral infiltrated brain tissue will be collected to investigate MR and GR and the effect of cortisol on these receptors.

### Concomitant care and trial follow-up

Patient care is continued as usual. Participants are allowed to participate in other trials if they are not interfering with our intervention period. Participants may withdraw from the study for any reason at any time. Premature study

withdrawal will have no effect on regular patient care. All data collected up to withdrawal will be used for analysis. All participants will be followed until completion of the trial and will be notified of the results.

### Primary outcome

The primary study outcome is the difference in proportion of patients scoring ≥3 points on at least one of the BPRS questions 5 days postoperatively – or earlier at discharge – between the hydrocortisone and placebo group. A better outcome on the BPRS in the hydrocortisone group would be considered a positive result, which would indicate that addition of hydrocortisone to dexamethasone is beneficial to reduce neuropsychiatric adverse effects of dexamethasone.

### Secondary outcomes

Secondary outcomes are neuropsychiatric symptoms, quality of sleep, HRQoL, and neurocognitive functioning 5 days postoperatively – or earlier at discharge –, 2 weeks, 5 to 8 weeks, and 3 months postoperatively. Furthermore, neuropsychiatric history, SAEs, prescribed (psychiatric) medication and referrals or evaluations of psychiatrist/psychologist will be collected. In addition, the laboratory measurements are considered secondary outcomes.

### Sample size

The sample size calculation is based on the results of a pilot study in which we tested the BPRS in our study population. We determined postoperative BPRS scores to evaluate the presence of psychiatric symptoms in brain tumour patients who underwent surgery and were treated with dexamethasone. We interviewed 18 patients on the day of hospital discharge. According to the BPRS scores in this group we considered a score of ≥3 on at least one of the BPRS items as clinically relevant for patients. We found that 7 out of 18 patients (≈40%) scored at least ≥3 on one of the BPRS items. Based on expert clinical judgement a 50% reduction in the patients with a score ≥3 is perceived as relevant. This equals an absolute difference of 20%. If we consider an α=0.05 and a power of 80%, and we take the 20% absolute difference between the two treatment arms, a sample size of 82 patients in each group would be needed. Based on an expected drop-out rate of ≈10%, 90 patients in each treatment arm (in total 180 patients) will be considered as a satisfactory sample size.

### Planned statistical analyses

This study is a randomised controlled trial. All data will be analysed on an intention-to-treat basis.

### BPRS

The primary study parameter is the proportion of patients scoring ≥3 points on at least one of the BPRS questions 5 days postoperatively – or earlier at discharge. The difference in these proportions between the placebo group and the hydrocortisone group will be calculated and statistically compared with the Chi-Square test. Any difference with a p<0.05 will be considered significant.

As secondary analyses, both treatment arms will be compared over time using analysis of repeated measures, generalised estimating equation (GEE) for the dichotomous outcomes as described in the primary outcome and the mixed linear model for the BPRS as continuous outcome. Additionally, we will look at the differences between the treatment arms on the separate items of the BRPS.

### Mini-screen, HADS, PANAS, ASRM, NPI-Q, DOS, LSEQ and HRQoL

Both treatment arms will be compared over time using analysis of repeated measures, GEE when outcome is dichotomous and mixed linear models for continuous outcomes.

Explanatory univariable and multivariable regression analysis will be used to identify (baseline) predictors of treatment response. Sensitivity analysis will be performed using multiple imputation to handle missing data. Any difference with a p<0.05 will be considered significant. A better outcome on the different scales in the hydrocortisone group would be considered a positive result.

### Neurocognitive functioning

The neurocognitive tests are a combination of neuropsychological tests for which normative data were published, so that patient's performance can be corrected according to age, gender and education. We will compare the raw data between the treatment arms with a t-test. Test scores can be transformed into z-values using normative data and compared with a one sample t-test. With no-normal distributions the appropriate statistical tests will be used.

### Subgroup analyses

Subgroup analyses will be performed to explore differences in our primary and secondary parameters for type of tumour, gender, age and dosage of dexamethasone.

As the study remains blinded until completion, no interim analyses are planned. In addition, the statistical plan will be finalised before starting the analyses.

### Data management

All data will be collected, stored, and analysed in a coded manner, and in accordance with the General Data Protection Regulation (GDPR). Data are collected and stored in the web-based database management system Castor EDC using electronic case report forms (CRFs). The CRFs have built-in data validity and value range checks. Source documents include medical records, paper CRFs, paper interviews, questionnaires and neuropsychological tests. All data is pseudonymised with a study code and will be stored for 15 years. The subject identification log which links subjects to the code is kept in a trial file only accessible to study personnel. Possible blood samples and residual tumour tissue obtained for standard care and obtained following local consent forms during the study will be stored according to local regulatory guidelines.

## Safety

Participating in the trial is very unlikely to bring additional risks for patients, since hydrocortisone is given in a low dose to restore the physiological cortisol level in the body.

### Data Safety Monitoring Board

This study established a DSMB that protects and guarantees patients safety during the trial. The DSMB includes three members with different study backgrounds: a statistician, psychiatrist and an internist/clinical epidemiologist. The role and responsibilities of the DSMB are described in the DSMB charter document.

### Adverse events and serious adverse events

The study population has a high risk of complications, which are inherent to their vulnerable condition (neurological deficits caused by a brain tumour, surgery, high-dose dexamethasone), and these are considered unrelated to the intervention. The patient population is expected to encounter many AEs not related to the experimental design; therefore, not all AE will be registered. AEs that will be registered are: psychiatric reports of which psychiatric consultation and medication is needed, brain oedema for which increased dexamethasone medication is given, or for which the dexamethasone reduction scheme is extended, all infections for which antibiotics are needed, cerebrospinal fluid leakage that requires surgical intervention, hyperglycaemia for which medication is needed, cardiovascular events requiring intervention, epileptic seizures requiring intervention and secondary adrenal insufficiency that requires hydrocortisone suppletion. The abovementioned AEs will be registered from the start of the intervention until completion of the trial.

SAEs that we will record are any untoward medical occurrence or effect that result in death, are life threatening (at the time of the event), require hospitalisation or prolongation of existing inpatients' hospitalisation — including intensive care — when a patient has unexpected longer IC admission or unexpected >10 days hospitalisation, result in persistent or significant disability or incapacity — like neurological dysfunction — lasting for more than 3 days postoperatively, result in any other important medical event not mentioned above due to medical or surgical intervention and judged as such by the investigators and postoperative haemorrhage for which recraniotomy is needed. An elective hospital admission will not be considered an SAE. Whether related to the study medication or not, all these SAEs occurring during the intervention period will be reported through the web portal governed and issued by the Dutch National Ethics Committee (The Central Committee on Research Involving Human Subjects (CCMO)) called 'ToetsingOnline'. SAEs occurring after the intervention period will be line listed and not reported in ToetsingsOnline.

### Monitoring

According to Good Research Practice, independent on-site monitoring visits will be performed to ensure that the conduct of the trial is consistent with the approved protocol(s), the rights and well-being of the subjects are protected, source documents are verified and data are accurately reported from the source documents. Monitoring visits will take place according to the study-specific monitoring plan.

### Patients and public involvement

Patients and public were not involved in the study design. The ZonMw committee that granted the funding includes a representative of the patients interests. The committee is kept informed annually.

## ETHICS AND DISSEMINATION

The study protocol has been approved by the accredited Medical Research Ethics Committee of the LUMC, by the Dutch competent authority and by the Institutional Review Boards of the participating sites. Substantial amendments to the protocol will first be sent for approval, and on consent, the participating centres and trial registries will be informed. The study will be conducted according to the principles of the Declaration of Helsinki, Good Clinical Practice guidelines — including centre-specific requirements — and in accordance with the Dutch Medical Research Involving Human Subjects Act (WMO). Participant WMO subject insurance is provided by the LUMC for all participating subjects and liability insurances are provided by all participating centres. This trial is registered in the Netherlands Trial Register (NTR). Regardless of the outcome, trial results will be submitted for publication in a peer-reviewed journal and presented at conferences. Publication rights and ownership of data are secured in a clinical trial agreement with participating centres. Full protocol, data sets and statistical analyses will be available in a repository with a persistent identifier after publication of the final results.

## TRIAL STATUS

The first patient was included on 10 September 2019 and recruitment is ongoing. Inclusion rate has slowed down due to the COVID-19 pandemic from March 2020 onwards. End of recruitment is expected to be 2022. The current study protocol version is V.3, 28 August 2020.

## DISCUSSION

The DEXA-CORT trial is the first double-blind, placebo-controlled trial in adult patients with a brain tumour, in which we investigate the possible prevention of neuropsychiatric adverse effects of perioperative dexamethasone treatment. Dexamethasone is a well-known very potent activator of the GR and consequently suppresses endogenous adrenocortical cortisol production via negative

feedback on the HPA-axis, depleting brain MR from its ligand.[12–14] The resultant disturbed balance in receptor activation in the brain may predispose for psychopathology. In this study, we test the hypothesis that reinstating brain MR activity via cotreatment with a physiological dose of cortisol (=hydrocortisone) will improve (specific features of) neuropsychological well-being. Because of the high dexamethasone doses and high affinity of dexamethasone for GR, cortisol treatment will only affect MR occupancy and not interfere with the therapeutic goal of dexamethasone treatment, which is GR activation.

Adverse neuropsychiatric effects can also be present with the use of other GCs, like prednisone and prednisolone. Like dexamethasone, prednisone and prednisolone suppress the HPA-axis, which would subsequently deplete MR from its ligand. In addition, compared with the naturally produced cortisol, the synthetic GCs dexamethasone, prednisone and prednisolone all have much lower MR binding potency.[11] An important difference between dexamethasone and the other mentioned GCs is its higher GR potency, its long half-life and lower MR potency. The potency differences could be of influence on the adverse effects; higher GR potency could suppress the HPA-axis faster, and longer half-life might extend its effects. Lower MR potency might accentuate the effect of a suppressed HPA-axis. It would be interesting to investigate the difference in neuropsychiatric effects between different GCs and (eventually) benefit of hydrocortisone add on.

This cortisol refill approach was already performed in childhood leukaemia patients. Patients who experienced severe adverse effects of dexamethasone seemed to benefit from the additional hydrocortisone medication.[21] To establish this potential positive effect in an adult population, the DEXA-CORT trial is investigating this principle in adult patients with a brain tumour. Dexamethasone is a standard clinical treatment in patients with a brain tumour perioperatively and adverse effects are also present in this patient group, which leads to reduced HRQoL. A methodological drawback of our study population is the lack of a cross-over design. Rather, the high doses, but brief perioperative use of dexamethasone, formed the rational for testing our hypothesis in this patient group. If additional hydrocortisone to dexamethasone proves effective in reducing its neuropsychiatric adverse effects, it would offer a new treatment option for this patient group and maybe even for many other patients who receive long-term treatment with dexamethasone and other synthetic GCs. An interesting question that will follow is how long to continue the hydrocortisone replacement. After dexamethasone treatment, the endogenous HPA-axis will need a recovery period. It might be beneficial to continue the additional hydrocortisone for at least some days after dexamethasone is stopped.

**Author affiliations**
[1]Department of Medicine, Division of Endocrinology, Leiden University Medical Center, Leiden, The Netherlands
[2]Department of Neurosurgery, Erasmus Medical Center, Rotterdam, The Netherlands
[3]Department of Psychiatry (GGZ inGeest), Amsterdam UMC (location VUmc), Vrije University, Amsterdam Public Health and Amsterdam Neuroscience Research Institutes, Amsterdam, The Netherlands
[4]Department of Anatomy and Neurosciences, Amsterdam UMC (location VUmc), Vrije University, Amsterdam, The Netherlands
[5]Department of Neurosurgery, University Neurosurgical Center Holland, Leiden University Medical Center, Haaglanden Medical Center and Haga Teaching Hospitals, Leiden and The Hague, The Netherlands
[6]Department of Medicine, Division of Endocrinology, and Centre for Endocrine Tumors Leiden (CETL), Leiden University Medical Center, Leiden, The Netherlands
[7]Department of Internal Medicine, Division of Endocrinology, Erasmus Medical Center, Rotterdam, The Netherlands

**Contributors** OCM was main applicant in the funding acquisition. A-SCAMK, DDS, AHZN, NRB, EFCvR, CHV, CMFD, AMP, WRvF and OCM participated in the design of the study. CMFD, DDS, AMP, WRvF and RDSNT are principal investigators at the participating study centres. CMFD, DDS, RDSNT, WAM, A-SCAMK initiated and implemented the study design at the participating study centre. A-SCAMK is the coordinating investigator with DDS as daily supervisor. A-SCAMK drafted the manuscript. All authors reviewed and approved the final manuscript.

**Funding** This work is supported by The Netherlands Organisation for Health Research and Development (ZonMw) and Dutch Brain Foundation grant project number 95 105 005. The DEXA-CORT study is investigator-initiated; Leiden University Medical Centre (LUMC) is the sponsor of the study. The funding party did play no role in the final study design or study conduct nor in the manuscript preparation of decision to submit for publication.

**Competing interests** None declared.

**Patient consent for publication** Not applicable.

**Provenance and peer review** Not commissioned; externally peer reviewed.

**ORCID iDs**
Anne-Sophie C A M Koning http://orcid.org/0000-0001-8809-2576
Christiaan H Vinkers http://orcid.org/0000-0003-3698-0744
Amir H Zamanipoor Najafabadi http://orcid.org/0000-0003-2400-2070
Nienke R Biermasz http://orcid.org/0000-0001-5817-3594
Elisabeth F C van Rossum http://orcid.org/0000-0003-0120-4913
Alberto M Pereira http://orcid.org/0000-0002-1194-9866
Wouter R van Furth http://orcid.org/0000-0001-5208-921X
Onno C Meijer http://orcid.org/0000-0002-8394-6859

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
