## [Reviewer comments · BMJ Open]

ARTICLE DETAILS

TITLE (PROVISIONAL)	The DEXA-CORT trial: study protocol of a randomized placebo-controlled trial of hydrocortisone in brain tumour patients on the prevention of neuropsychiatric adverse effects caused by perioperative dexamethasone
AUTHORS	Koning, Anne-Sophie; Satoer, Djaina; Vinkers, C; Zamanipoor Najafabadi, Amir H.; Biermasz, Nienke; Nandoe Tewarie, Rishi D.S.; Moojen, Wouter A.; van Rossum, Elisabeth; Dirven, Clemens; Pereira, Alberto; van Furth, W.R.; Meijer, Onno C.

VERSION 1 – REVIEW

REVIEWER	Lightman, Stafford University of Bristol
REVIEW RETURNED	16-Sep-2021

GENERAL COMMENTS	This is an important study which should have been performed long ago. I congratulate the authors for getting the money to do this overdue study.
--

REVIEWER	Spencer-Segal, Joanna L University of Michigan Molecular and Behavioral Neuroscience Institute
REVIEW RETURNED	20-Sep-2021

GENERAL COMMENTS	In this paper, the authors present the study protocol of the DEXA-CORT trial, designed to determine whether hydrocortisone at physiological doses, when added to dexamethasone treatment perioperatively for brain tumor resection in adult patients, decreases psychiatric adverse events. The rationale for this is that the removal of physiologic MR stimulation by high doses of dexamethasone leads to neuropsychiatric adverse events which may be ameliorated by adding physiologic cortisol. The protocol is generally clearly presented and contains most of the necessary details. There are several considerations or additions to suggest prior to publication: 1. The authors present, as part of the rationale, that adverse neuropsychiatric events are frequently experienced with dexamethasone therapy. Yet, they are experienced with many other glucocorticoids as well including prednisolone and prednisone. As part of the rationale, it would be nice to have more consideration of the basis for thinking that MR depletion is important for these adverse effects. To the extent that they may be more frequent with dexamethasone, could that be explained by its higher potency at the GR and its long half-life?
--

	2. Neurosurgery is something of a “planned critical illness” similar to some cardiac surgery. PTSD is common in these patients and glucocorticoids modulate PTSD risk. A specific PTSD questionnaire would have been interesting to see the effect of hydrocortisone on this outcome – not likely to be added now, but this reviewer was curious if this was considered/why it was not included. 3. Is there a hypothesis about what will happen to symptoms in the ‘withdrawal’ phase after dex +/- hydrocortisone is stopped? Hydrocortisone is not planned to continue through this phase and yet, some recovery period for the endogenous HPA axis is expected after cessation of dexamethasone treatment even after a taper. Also, dex has a long half-life and so if the last day for administering dex and hydrocortisone is the same, the dexamethasone will last longer, and then the HPA axis will take some time to recover. During this time the participants would be relatively depleted of MR stimulation. Was there consideration of continuing hydrocortisone for a time past the cessation of dexamethasone? 4. The prior findings in pediatric patients with ALL form a strong rationale to try this in adults. However, while the pediatric study was a crossover design, in this study, two randomized groups will be compared. These are adult patients with brain tumors and undergoing brain surgery which could present a major confound in terms of interpreting neuropsychiatric outcomes (these are expected to be influenced by the underlying pathology including, importantly, the location of the tumor and extent of the surgery). More discussion of how this was considered in the trial design is welcomed. 5. Figure 1 illustrates the HPA aspect of GR/MR stimulation but does not well illustrate that the problem under consideration – potentially solved by giving hydrocortisone – is lack of MR stimulation in higher brain areas controlling mood, cognition, etc. 6. Were patients with prior diagnosis of secondary adrenal insufficiency/already taking hydrocortisone excluded? Presumably and this should be indicated in the figure. 7. Do the participants receive instructions about what time of day to take the hydrocortisone? Presumably morning and early afternoon to mimic diurnal rhythm but it is not specified.
--	--

VERSION 1 – AUTHOR RESPONSE

Reviewer: 1

Dr. Stafford Lightman, University of Bristol

Comments to the Author:

This is an important study which should have been performed long ago. I congratulate the authors for getting the money to do this overdue study.

Changes in the manuscript:

We thank the reviewer for this very positive remark. No changes made in the manuscript.

Reviewer: 2

Dr. Joanna L Spencer-Segal, University of Michigan Molecular and Behavioral Neuroscience Institute

Comments to the Author:

In this paper, the authors present the study protocol of the DEXA-CORT trial, designed to determine whether hydrocortisone at physiological doses, when added to dexamethasone treatment perioperatively for brain tumor resection in adult patients, decreases psychiatric adverse events. The rationale for this is that the removal of physiologic MR stimulation by high doses of dexamethasone leads to neuropsychiatric adverse events which may be ameliorated by adding physiologic cortisol.

The protocol is generally clearly presented and contains most of the necessary details. There are several considerations or additions to suggest prior to publication:

1. The authors present, as part of the rationale, that adverse neuropsychiatric events are frequently experienced with dexamethasone therapy. Yet, they are experienced with many other glucocorticoids as well including prednisolone and prednisone. As part of the rationale, it would be nice to have more consideration of the basis for thinking that MR depletion is important for these adverse effects. To the extent that they may be more frequent with dexamethasone, could that be explained by its higher potency at the GR and its long half-life?

We thank the reviewer for the positive and very constructive comments. We are not aware of differences in adverse neuropsychiatric events between different glucocorticoids. We certainly agree that that potency could be a factor of influence; higher GR potency leads to a more rapid suppression of the HPA-axis and longer half-life might extend the effects. However, it needs to be mentioned that the dosages used of synthetic GCs most often exceed physiological levels by a factor of > 10, resulting in virtually complete inhibition of the HPA-axis. Compared to our naturally produced cortisol, dexamethasone, prednisone and prednisolone all have (much) lower MR potency. And according to the literature, low MR activity seems to be associated with psychopathology. And despite their difference in GR potency, the mentioned glucocorticoids all very potently suppress the production of cortisol, which subsequently deplete MRs from its ligand. However, prednisolone may be an 'in between' ligand with higher residual MR potency/efficacy than most other synthetic GCs. In case of a positive outcome of our trial, it will be very interesting evaluating hydrocortisone add on in relation to predniso(lo)ne.

Changes in the manuscript:

Discussion, Lines 422 – 431, we added: Adverse neuropsychiatric effects can also be present with the use of other GCs, like prednisone and prednisolone. Like dexamethasone, prednisone and prednisolone suppress the HPA-axis, which would subsequently deplete MR from its ligand. In addition, compared to the naturally produced cortisol, the synthetic GCs dexamethasone, prednisone, and prednisolone all have (much) lower MR potency (11). An important difference between dexamethasone and the other mentioned GCs is its higher GR potency, its long half-life, and lower MR potency. The potency differences could be of influence on the adverse effects; higher GR potency could maybe suppress the HPA-axis faster and longer half-life might extend its effects. Lower MR potency might accentuate the effects of a suppressed HPA-axis. It would be interesting to investigate the difference in neuropsychiatric effects between different GCs, and (eventually) benefit of hydrocortisone add on.

2. Neurosurgery is something of a “planned critical illness” similar to some cardiac surgery. PTSD is common in these patients and glucocorticoids modulate PTSD risk. A specific PTSD questionnaire would have been interesting to see the effect of hydrocortisone on this outcome – not likely to be added now, but this reviewer was curious if this was considered/why it was not included.

We thank the reviewer for this valuable remark. Our focus was the neuropsychiatric effects that could occur during the use of glucocorticoids, but we did not consider evaluating PTSD specifically, which would have been interesting indeed.

Changes in the manuscript:

Lines 196 – 199, We added: We plan to evaluate potential long term follow-up effects on health-related quality of life (HRQoL). However, we did not include a specific measure for PTSD, even if a 'planned critical illness' like elective brain surgery may induce this disease (24).

3. Is there a hypothesis about what will happen to symptoms in the 'withdrawal' phase after dex +/- hydrocortisone is stopped? Hydrocortisone is not planned to continue through this phase and yet, some recovery period for the endogenous HPA axis is expected after cessation of dexamethasone treatment even after a taper. Also, dex has a long half-life and so if the last day for administering dex and hydrocortisone is the same, the dexamethasone will last longer, and then the HPA axis will take some time to recover. During this time the participants would be relatively depleted of MR stimulation. Was there consideration of continuing hydrocortisone for a time past the cessation of dexamethasone?

The decision to stop hydrocortisone at the same time as dexamethasone is a practical one; it is very clear both for the patients and caretakers. A lot is going on and this way the burden is the lowest for the patients and chances of mistakes in medication are minimized.

If this intervention proves to have beneficial effects, we agree with the reviewer that continuation of hydrocortisone replacement could well have additional beneficial effects for at least some days.

Changes in the manuscript:

Lines 442 - 446, We added: An interesting question that will follow is how long to continue the hydrocortisone replacement. After dexamethasone treatment, the endogenous HPA-axis will need a recovery period. It might be beneficial to continue the additional hydrocortisone for at least some days after dexamethasone is stopped.

4. The prior findings in pediatric patients with ALL form a strong rationale to try this in adults. However, while the pediatric study was a crossover design, in this study, two randomized groups will be compared. These are adult patients with brain tumors and undergoing brain surgery which could present a major confound in terms of interpreting neuropsychiatric outcomes (these are expected to be influenced by the underlying pathology including, importantly, the location of the tumor and extent of the surgery). More discussion of how this was considered in the trial design is welcomed.

We completely agree with the reviewer that neuropsychiatric outcomes in these patients is also affected by other factors that are an intrinsic part of the underlying disease and its treatment. This is why randomisation in this study is imperative. With the randomisation procedure we intend to equally distribute those factors across the study arms to cancel out these factors. We do collect information such as the underlying pathology and on the location of the tumor.

Changes in the manuscript:

Lines 114 - 116, We added: With randomisation we intend to equally distribute factors that can have an influence on our main outcome; neuropsychiatric effects. For example, the underlying pathology, the location of the tumour, extent of the surgery, and psychiatric history.

Lines 437 – 439, We added: A methodological drawback of our study population is the lack of a cross-over design. Rather, the high doses, but brief perioperative use of dexamethasone formed the rationale for testing our hypothesis in this patient group.

5. Figure 1 illustrates the HPA aspect of GR/MR stimulation but does not well illustrate that the problem under consideration – potentially solved by giving hydrocortisone – is lack of MR stimulation in higher brain areas controlling mood, cognition, etc.

We agree, and adjusted Figure 1 accordingly.

Changes in the manuscript:

Figure 1 is adjusted by now also highlighting the MRs in higher brain areas.

6. Were patients with prior diagnosis of secondary adrenal insufficiency/already taking hydrocortisone excluded? Presumably and this should be indicated in the figure.

Yes, however we realised this is not (yet) included in the exclusion criteria. We added this exclusion criterium in figure 2.

Changes in the manuscript:

Figure 2, We added to exclusion criteria: Patients with prior diagnosis of secondary adrenal insufficiency/already taking hydrocortisone.

7. Do the participants receive instructions about what time of day to take the hydrocortisone? Presumably morning and early afternoon to mimic diurnal rhythm but it is not specified.

Yes, during their stay in the hospital the hydrocortisone/placebo is given by the nurses. It is registered in the electronic registry and the nurses have to digitally sign off the administration. It is administered in the morning at 08:00hr and in the evening between 18:00hr and 20:00hr, because at these times dexamethasone is administered according to local standard procedures. To minimise the number of errors in medicine administrations we adhered to these existing logistics. At hospital discharge, they receive a dexamethasone taper schedule. Depending on the schedule, they receive further instructions on the hydrocortisone/placebo administration (both verbally and on paper). For manageability reasons we decided to always take the hydrocortisone/placebo at the first and the last time point of the dexamethasone administration.

Changes in the manuscript:

Lines 170 - 180, We added: During hospitalisation the hydrocortisone or placebo is administered in the morning at 08:00hr and in the evening between 18:00hr and 20:00hr, because at these times dexamethasone is administered according to local standard procedures. To minimise the number of errors in medicine administrations we decided to adhere to the existing logistics. When participants go home and have to taper the dexamethasone, they receive a taper schedule from the nurses. Depending on the schedule, they receive further instructions on the hydrocortisone/placebo administration (both verbally and on paper). For manageability reasons we decided to always take the hydrocortisone/placebo at the first and the last time point of the dexamethasone administration. Study medication and information is handed out to the patients by the nurses, in consultation with the coordinating investigator. Patients will receive the amount of study medication they need to minimise medication errors. Drug accountability is checked by the coordinating investigator.

VERSION 2 – REVIEW

REVIEWER	Spencer-Segal, Joanna L University of Michigan Molecular and Behavioral Neuroscience Institute
REVIEW RETURNED	02-Nov-2021
GENERAL COMMENTS	All my comments have been addressed.